# Novel Substituted Azoloazines with Anticoagulant Activity

**DOI:** 10.3390/ijms242115581

**Published:** 2023-10-25

**Authors:** Alexander A. Spasov, Olga V. Fedorova, Nikolay A. Rasputin, Irina G. Ovchinnikova, Rashida I. Ishmetova, Nina K. Ignatenko, Evgeny B. Gorbunov, Gusein A. o. Sadykhov, Aida F. Kucheryavenko, Kseniia A. Gaidukova, Victor S. Sirotenko, Gennady L. Rusinov, Egor V. Verbitskiy, Valery N. Charushin

**Affiliations:** 1Department of Pharmacology & Bioinformatics, Scientific Center for Innovative Drugs, Volgograd State Medical University, Volgograd 400131, Russia; aidakucheryavenko@yandex.ru (A.F.K.); ksenijagajjdukva@rambler.ru (K.A.G.); post@vsirotenko.ru (V.S.S.); 2I. Ya. Postovsky Institute of Organic Synthesis, Ural Branch of the Russian Academy of Sciences, Ekaterinburg 620108, Russia; fedorova@ios.uran.ru (O.V.F.); iov@ios.uran.ru (I.G.O.); iri@ios.uran.ru (R.I.I.); nki@ios.uran.ru (N.K.I.); nitro@ios.uran.ru (E.B.G.); gusein@ios.uran.ru (G.A.o.S.); rusinov@ios.uran.ru (G.L.R.); verbitskye@yandex.ru (E.V.V.); valery-charushin-562@yandex.ru (V.N.C.); 3Department of Organic and Biomolecular Chemistry, Ural Federal University Named after the First President of Russia B. N. Yeltsin, Ekaterinburg 620002, Russia; 4Department of Technology & Organic Synthesis, Ural Federal University Named after the First President of Russia B. N. Yeltsin, Ekaterinburg 620002, Russia

**Keywords:** azolo[1,5-*a*]pyrimidines, indolo[2,3-*b*]quinoxalines, 1,2,4-triazolo[5,1-*c*]triazines, azolo[1,2,4,5]tetrazines, anticoagulant, cytokine storm

## Abstract

Hypercytokinemia, or cytokine storm, often complicates the treatment of viral and bacterial infections, including COVID-19, leading to the risk of thrombosis. However, the use of currently available direct anticoagulants for the treatment of COVID-19 patients is limited due to safety reasons. Therefore, the development of new anticoagulants remains an urgent task for organic and medicinal chemistry. At the same time, new drugs that combine anticoagulant properties with antiviral or antidiabetic activity could be helpfull in the treatment of COVID-19 patients, especially those suffering from such concomitant diseases as arterial hypertension or diabetes. We have synthesized a number of novel substituted azoloazines, some of which have previously been identified as compounds with pronounced antiviral, antibacterial, antidiabetic, antiaggregant, and anticoagulant activity. Two compounds from the family of 1,2,4-triazolo[1,5-*a*]pyrimidines have demonstrated anticoagulant activity at a level exceeding or at least comparable with that of dabigatran etexilate as the reference compound. 7,5-Di(2-thienyl)-4,5-dihydro-[1,2,4]triazolo[1,5-*a*]pyrimidine has shown the highest ability to prolong the thrombin time, surpassing this reference drug by 2.2 times. This compound has also exhibited anticoagulant activity associated with the inhibition of thrombin (factor IIa). Moreover, the anticoagulant effect of this substance becomes enhanced under the conditions of a systemic inflammatory reaction.

## 1. Introduction

Anticoagulant drugs are widely used for the treatment and prevention of thrombosis in case of patients with sepsis of various etiologies. However, modern direct oral anticoagulants are not always compatible with new antiviral agents, as has recently been established in the treatment of COVID-19 patients [1]. Therefore, the development of novel anticoagulants remains an urgent task for organic and medicinal chemistry. Drugs that combine anticoagulant properties with antiviral or antibacterial activity [2] are of particular interest. In this regard, substituted azoloazines, also exhibiting antidiabetic activity as a decisive factor in their use for treatment of COVID-19 and other viral and bacterial infections [3], comprise a promising family of biologically active compounds. Intra- or extracellular accumulation of glycation end products is considered to be an important factor for the pathogenesis of such diseases as atherosclerosis, rheumatoid arthritis, inflammation, atherosclerosis, and heart failure [4]. The latter is directly related to the hematopoietic system, in particular, with an increased formation of platelets.

Azoloazines are azaheterocyclic compounds containing fused five- and six-membered rings. These compounds can be regarded as structural analogues of purine bases. Indeed, these compounds can act as antimetabolites or enzyme inhibitors, and are known to have a wide spectrum of biological activity [5,6] (Figure 1). In particular, they exhibit antiviral activity, and some compounds of this series are characterized by activity against the main 3CLpro-2 protease of the coronavirus, which is involved in the maturation and replication of the SARS-CoV-2 virus [7]. In addition, azoloazines have demonstrated a pronounced protective effect against septic shock and exhibit antibacterial, antidiabetic, and anticoagulant activities [8,9,10]. It is worth mentioning that substituted 1,2,4-triazolo[1,5-*a*]pyrimidin-7-ones have a higher anticoagulant activity than that of the well-known drug dabigatran etexilate, which also contains azole and azine moieties [2].

We have found polyvalent antidiabetic activity in one of these derivatives [11], and have shown that 7-(4-methoxyphenyl)-5-phenyl-4,5-dihydro-[1,2,4]triazolo[1,5-*a*]pyrimidine appears to be a glucokinase activator and dipeptidyl peptidase type 4 inhibitor. Also, we have identified amino derivatives of 5,7-diaryl substituted [1,2,4]triazolo[1,5-*a*]pyrimidines, which are micromolar inhibitors of IL-6 secretion and NO synthesis, capable of preventing LPS-induced acute lung injury [12]. Previously, a series of substituted azolo[1,2,4,5]tetrazines have been synthesized, including compounds which have exhibited pronounced antibacterial and antiglycating activities [13]. Considering all these data, it appears to be worth testing analogues of the mentioned azoloazines as potential anticoagulants.

In this communication, we wish to describe new derivatives of the azoloazine family with varied numbers and positions of nitrogen atoms in the azine moiety (two, three, or four atoms), and in the azole fragment (one, two, or three atoms) as potential anticoagulants. These compounds involve 6-(2-Alkyl)-6*H*-indolo[2,3-*b*]quinoxalines, substituted azolo[1,5-*a*]pyrimidines, and 1,2,4-triazolo[5,1-*c*]triazines, as well as azolo[1,2,4,5]tetrazine derivatives.

## 2. Results and Discussion

### 2.1. Chemistry

Substituted indolo[2,3-*b*]quinoxalines are known to be DNA-intercalating antiviral agents [14] and can also exhibit antidiabetic activity [15]. It is interesting to study the anticoagulant activity of substituted indolo[2,3-*b*]quinoxalines. Therefore, a new series of indolo[2,3-*b*]quinoxaline derivatives **1a**–**j** have been obtained from 2-(2-bromophenyl)quinoxalines through a microwave-assisted Buchwald−Hartwig cross-coupling reaction, followed by intramolecular oxidative cyclodehydrogenation (Figure 1) [16].

Substituted [1,2,4]triazolo[1,5-*a*]pyrimidines, the structural isomers of natural purines, belong to another important class of nitrogen-containing heterocycles which has received great attention during recent decades due to its wide application in pharmaceuticals [17]. They have been proven to exhibit antiviral [18], antibacterial [19], tuberculostatic [20], and antitumor [21] activities. In addition, there are references in the literature indicating that compounds based on the [1,2,4]triazolo[1,5-*a*]pyrimidine core appear to be anti-inflammatory agents [22].

To find out the “structure–activity” relations for substituted azolo[1,5-*a*]pyrimidines, we synthesized compounds bearing aryl, hetaryl or the nitro substituents in the pyrimidine ring [6]. Hydrogenated analogues of azolo[1,5-*a*]pyrimidines, as well as their salts with guanidine, which are an important fragment of the known anticoagulants, have been obtained.

The following approaches have been used for the synthesis of new representatives of 5,7-di(hetero)aryl-substituted [1,2,4]triazolo[1,5-*a*]pyrimidines. Compounds **5a**–**d** have been synthesized using the so-called S_N_^H^ methodology (nucleophilic substitution of hydrogen) [23,24,25]. Bromination of [1,2,4]triazolo[1,5-*a*]pyrimidine has produced bromo compound **2**; its further reaction with a Grignard reagent led to the formation of intermediate σ^H^-adducts **3a**,**b**, which on treatment with triethylamine (TEA) were converted into aromatic compounds **4a**,**b** (Figure 2). Subsequent functionalization at C-5 position of compounds **4a**,**b** with a second Grignard reagent afforded the target 5,7-di(hetero)aryl-substituted-4,5-dihydro-[1,2,4]triazolo[1,5-*a*]pyrimidines **5a**–**d** [26].

The three-step synthesis of amino derivatives of di(hetero)aryl-substituted azolo[1,5-*a*]pyrimidines **7a**,**b** from commercially available reagents is shown in Figure 2. The interaction of aromatic aldehydes with aromatic ketones results in the formation of the corresponding nitrochalcones **6a**,**b** [27,28]. These nitrochalcones react with aminoazoles to produce nitro derivatives of di(hetero)aryl-substituted azolo[1,5-*a*]pyrimidines **4c**,**d**. Amino derivatives of di(hetero)aryl-substituted azolo[1,5-*a*]pyrimidines **7a**,**b** have been synthesized by reduction of the nitro group in the presence of hydrazine hydrate and Raney nickel.

Dihydro derivatives of [1,2,4]triazolo[1,5-*a*]pyrimidines **10a**,**b** have been obtained from compounds **8a**,**b** derived from a three-component Biginelli reaction [29], while compound **8c** could not be subjected to further modification. Amino derivatives **8a**,**b** have been condensed with Boc-protected L-glycine, with TBTU as a coupling agent (Figure 2).

Guanidinium salts of 6-nitro1,2,4-triazolo[1,5-*a*]pyrimidines **12a**,**b** and 6-nitro-1,2,4-triazolo[5,1-*c*]-triazine **14** have been synthesized through the reaction of the corresponding 6-nitro-1,2,4-triazoloazines **11a**,**b** or **13** with guanidine in methanol (Figure 3).

In addition to azoloazines containing two and three nitrogen atoms in the azine fragment, a number of substituted azolo[1,2,4,5]tetrazines have also been synthesized. It should be emphasized that compounds bearing the 1,2,4,5-tetrazine ring can exhibit antiaggregant activity [30].

Imidazo[1,2-*b*][1,2,4,5]tetrazines and [1,2,4]triazolo[4,3-*b*][1,2,4,5]tetrazines containing guanidine residue have been synthesized. The latter appear to be an important structural fragment providing anticoagulant activity to FXIa inhibitors [31]. The interaction of imidazo[1,2,4,5]tetrazine with free guanidine takes place in a mixture of acetonitrile and methanol without heating, thus affording 3-guanidino-6-alkylthioimidazo[1,2-*b*][1,2,4,5]tetrazines **15a,b** (Figure 4). 3-Guanidino-6-alkylthio-imidazo[1,2-*b*][1,2,4,5]tetrazines **16a–c** have been obtained in a similar way (Figure 5) [17].

### 2.2. Biology

#### 2.2.1. Anticoagulant Activity of the Target Compounds In Vitro

The effect of these compounds on coagulogram parameters was determined using in vitro experiments. We observed that dabigatran etexilate at a concentration of 100 μM increased the activated partial thromboplastin time (APTT) by 1.6 times relative to the control (Table 1). Dabigatran etexilate in the studied concentration increased the thrombin time (TT) by 6.0 times, which corresponds to the mechanism of its anticoagulant action—the disruption of the final stage of coagulation without changing the prothrombin time.

In the similar way, our study of targeted novel azoloazine derivatives was carried out with an emphasis on their effect on the coagulogram parameters of rabbit blood. The results are summarized in Table 1. It was shown that compound **5d** increased the APTT by 1.3 times relative to the control and had the greatest ability to prolong the thrombin time, exceeding the comparison drug dabigatran etexilate by 13.0 times. Compound **7a** was comparable to dabigatran etexilate in terms of activity. Compounds **1f**, **5a**, and **10a** prolonged the thrombin time by 3.5 times relative to the control. Other substances also significantly prolonged the thrombin time relative to control, but to a lesser extent than the comparison drug. None of the studied compounds affected the prothrombin time.

In this regard, compound **5d**, which demonstrated the greatest antithrombin activity in the in vitro experiment, was investigated further for its effect on the coagulogram parameters of rabbit blood treated with LPS to mimic the conditions of hypercytokinemia (Table 2). The coagulation parameters of blood treated with LPS changed, with the exception of PT. The APTT was lengthened by 1.2 times and TT was decreased by 1.3 times compared to the intact blood sample. At the same time, the reference drug dabigatran etexilate reliably prolonged the APTT and TT by 2.3 and 14.0 times, respectively. Compound **5d** had a significant effect only on the thrombin time, reliably exceeding the control values of LPS-treated blood by 17.0 times.

Due to the high antithrombin effect of the compound and reference drug, they were studied in a concentration range from 10 to 0.1 μM to calculate EC_50_ both in intact and LPS-treated PPP.

Thus, the comparison drug dabigatran etexilate at a concentration of 10 μM significantly prolonged this index up to 46.8 s, which corresponds to a reliable increase of 302.0% relative to the control. When the concentration was further reduced to 5 and 1 μM, the thrombin time was also statistically significantly prolonged by 176.1 and 42.3%, respectively. At a concentration of 0.1 μM, dabigitran etexilate had no effect on this parameter. The EC_50_ antithrombin activity of the comparison drug was 1.4 μM.

Compound **5d** at a concentration of 10 μM significantly prolonged the thrombin time up to 467.1% more than the control values. Further reductions to 5 and 1 μM resulted in differences relative to the control of 194.1 and 46.5%, respectively. At a concentration of 0.1 μM, compound **5d** had no effect on this parameter. The EC_50_ of antithrombin activity was equal to 1.25 μM.

Effect of compound **5d** and the comparison drug on coagulogram parameters were studied under conditions of hypercytokinemia in vitro. The thrombin time index in control samples of blood treated with lipopolysaccharide amounted to 9.0 s, which is significantly lower than these data in the intact blood group (Table 2). Dabigatran etexilate at a concentration of 10 μM significantly prolonged this index up to 292.0% relative to the control (Table 3). When the concentration was further reduced to 5 and 1 μM, the thrombin time was also significantly prolonged by 208.6 and 51.7%, respectively. At a concentration of 0.1 μM, dabigitran etexilate had no effect on this parameter. The EC_50_ of antithrombin activity of the comparison drug under conditions of hypercytokinemia was 0.76 μM (Table 3).

Compound **5d** at a concentration of 10 μM significantly prolonged the thrombin time index up to 354.1% relative to the control. When reducing the concentration of the substance to 5 and 1 μM, the thrombin time statistically significantly increased by 234.1 and 56.3%, respectively, relative to control values. At a concentration of 0.1 μM, compound **5d** did not affect this parameter. The EC_50_ of antithrombin activity under conditions of hypercytokinemia was equal to 0.78 μM (Table 3).

Thus, under conditions of hypercytokinemia, compound **5d** and dabigatran etexilate were comparable in terms of EC_50_ of antithrombin activity.

#### 2.2.2. Ecarin Clotting Time

The ecarin time of PPP control samples was 39.95 ± 0.96. The study of PPP with the addition of the comparison drug dabigatran etexilate at a concentration of 100 μM resulted in a 2.3-fold lengthening of the time of stable clot formation compared to control values. In another step of the experiment, the concentration of dabigatran etexilate was reduced to 10 and 1 μM, and as a result, there was a linear shortening of the studied index to 54.1 and 41.6 s, respectively.

Compound **5d** with an initial concentration of 100 μM showed a prolongation of the clotting time to 81.3 secs. At concentrations of 10 and 1 μM, there was a change in the clotting time to 50.5 and 40.2 secs, respectively.

Based on the data obtained, the EC_50_ was calculated. The EC_50_ of the comparison drug was 10.5 μM, while the EC_50_ of the compound under study was 17.9 μM. As a result of this study, the direct antithrombin effect of compound **5d** was shown to be comparable with the comparison drug dabigatran etexilate in terms of EC_50_ (Figure 2).

On the basis of the obtained data, it is possible to state that both of the tested compounds exhibit a pronounced direct antithrombin action.

#### 2.2.3. An Animal Study of Anticoagulant Activity

Dabigatran etexilate showed activity in blood for both group of intact animals, having hypercytokinemia and not. Thus, compound **5d** was administrated intragastricaly to these both groups of intact animals in a series of experiments in amount equal to dabigatran etexilate. Parameters of the coagulograms obtained in experiments using animal blood are presented in Table 4.

In experiments on intact control rats, the APTT was 38.3 s (Table 4). At the same time, dabigatran etexilate at a dose of 12 mg/kg significantly prolonged this indicator by 3.6 times within 2 h after intragastric administration. Compound **5d** at a dose equimolar to dabigatran etexilate of 5.5 mg/kg significantly increased this indicator compared to the control by 1.5 times within 2 h after administration.

The thrombin time of the blood of the intact control group of rats was 57.7 min. The reference drug dabigatran etexilate and compound **5d** increased this indicator by 10.5 and 5.6 times, respectively.

The prothrombin time did not change under the influence of the test substance and dabigatran etexilate (Table 4).

Coagulogram parameters in control blood samples changed significantly under conditions of a systemic inflammatory reaction. The APTT decreased by 2.1 times compared to intact control animals, and the thrombin time also decreased by 1.3 times, indicating the activation of coagulation hemostasis in rats.

Dabigatran etexilate significantly prolonged the APTT in the blood of LPS-injected rats, increasing it by 2.2 times in relation to the control group. At the same time, the thrombin time statistically significantly increased by 12.8 times. Compound **5d** also significantly prolonged the APTT by 1.6 times. At the same time, the thrombin time in the rat blood increased 14.5 times.

However, neither the test compound nor dabigatran etexilate had an effect on the prothrombin time. Thus, under conditions of a systemic inflammatory reaction, compound **5d** showed a more pronounced antithrombin effect than in its absence, which indicated the possible effect of this substance on the inflammatory components of blood (Table 4).

## 3. Materials and Methods

### 3.1. Chemistry

Commercial reagents were obtained from Sigma-Aldrich (St. Louis, MO, USA), Acros Organics (Waltham, MA, USA), or Alfa Aesar (Haverhill, MA, USA) and used without any further purification. All workup and purification procedures were carried out using analytical-grade solvents. ^1^H NMR spectra were recorded in DMSO-*d*_6_ solution on “Bruker DRX-400” devices using TMS and DMSO-*d*_6_ as an internal standard. The following abbreviations were used for NMR signals: s—singlet, d—doublet, t—triplet, q—quartet, dd—double doublet, m—multiplet, br—broaded. Electrospray ionization mass spectra were recorded for positive ions on a qTOF maXis Impact HD ultra-high resolution mass spectrometer (Bruker Daltonics, Billerica, MA, USA), with a standard ionization source in the mass range 50–2500 Da, by injection analysis for sample solutions in acetonitrile using a syringe pump inlet (model No. 601553 kdScientific Inc., Holliston, MA, USA); solution infusion rate of 240 µL/h) using a modified “Direct_Infusion 100–1000” preset method. Calibration of the mass scale was external, according to the signals of a lithium acetate solution using HPC or improved quadratic methods. All data were collected and processed using the Compass for oTof series 1.7 software package (oTOF Control 3.4; Bruker Compass DataAnalysis 4.2). IR spectra were recorded on a Spectrum One FT-IT spectrometer (Perkin Elmer, Waltham, MA, USA) in a range of 4000–400 cm^−1^ using a diffuse reflectance attachment. Elemental analysis was performed on a PerkinElmer PE 2400 elemental analyzer. Melting points were determined on a Stuart SMP3 and were uncorrected. The monitoring of reaction progress was performed using TLC on Silufol UV254 plates. Column chromatography was performed on Chromagel (silica gel, 400 mesh).

Compounds **1a**–**j**, **5a**–**d**, **8a**,**b**, **9a**,**b**, **8c**, and **15a**,**b** were synthesized in accordance with literature data: **1a**–**j** [16], **5a**–**d** [26], **8a**,**b**, **9a**,**b** [12], **8c** [29], and **15a**,**b** [30]. All synthesized compounds were >96% pure by elemental analysis. ^1^H and ^13^C NMR spectra of compounds **7a**,**c**, **10a**,**c**, **12a**,**c**, **14**, and **16a**–**c** are given in the Appendix A.

#### 3.1.1. General Procedure for the Preparation of Nitro Derivatives of Azolo[1,5-*a*]pyrimidines **4c**,**d** (Figure 2)

K_2_CO_3_ (1.0 mmol) was added to chalcone solution **1** (0.7 mmol), obtained by method [5] or [32] for **1a** or **1b**, and to aminoazole (1.0 mmol) in 10 mL of dimethylformamide. The mixture was stirred at 100 ℃ for 32 h. Water was added to the mixture upon completion of the process, and the formed precipitate was filtered. The product, eluting with chloroform, was purified chromatographically on a preparative column (SiO_2_) and crystallized from CH_3_CN:

5,7-*Bis*(4-nitrophenyl)imidazo[1,2-*a*]pyrimidine **4c**: light yellow solid; mp = 334–336 °C. Yield: 200 mg (83%). ^1^H NMR (400 MHz, DMSO-*d*_6_) δ, 8.61 (m (AA’BB’), 2 H, phenyl), 8.81 (m (AA’BB’), 2 H, phenyl), 8.41 (m (AA’BB’), 2 H, phenyl), 8.25 (m (AA’BB’), 2 H, phenyl), 8.05 (s, 1 H, CH, imidazopyrimidine), 8.00 (d, 1 H, *J* = 1.4 Hz, imidazopyrimidine), 7.94 (d, 1 H, *J* = 1.4 Hz, imidazopyrimidine). ^13^C NMR (101 MHz, DMSO-*d*_6_) δ 152.93, 148.73, 148.67, 148.45, 144.27, 142.58, 137.73, 136.79, 130.24, 128.41, 124.23, 123.97, 110.35, 106.74. HRMS(ESI): calculated for C_18_H_12_N_5_O_4_ [M + H]^+^: 362.0884, found: 362.0885; IR ν_max_ (cm^−1^): 3145, 3105, 3073, 1601, 1587, 1515, 1483, 1340, 1278, 1255, 1140, 1102, 8492, 830, 754, 732, 696. Anal. calcd. for C_18_H_11_N_5_O_4_: C, 59.84; H, 3.07; N, 19.38. Found: C, 59.87; H, 3.05; N, 19.40.

5-(4-Nitrophenyl)-7-(thiophen-2-yl)-[1,2,4]triazolo[1,5-*a*]pyrimidine **4d**: light yellow solid; mp = 246–247 °C. Yield: 210 mg (84%). ^1^H NMR (400 MHz, DMSO-*d*_6_) δ, 8.89 (s, 1 H, CH, triazolopyrimidine), 8.85 (dd, 1 H, *J* = 1.0, 4.0 Hz, thienyl), 8.69 (m (AA’BB’), 2 H, phenyl), 8.66 (s, 1 H, CH, triazolopyrimidine), 8.45 (m (AA’BB’), 2 H, phenyl), 8.26 (dd, 1 H, *J* = 1.0, 4.9 Hz, thienyl), 7.49 (dd, 1 H, *J* = 4.0, 4.9 Hz, thienyl). ^13^C NMR (101 MHz, DMSO-*d*_6_) δ 157.63, 156.38, 155.39, 148.84, 142.10, 141.79, 136.22, 133.77, 129.83, 129.07, 128.21, 123.93, 103.49. HRMS(ESI): calculated for C_15_H_10_N_5_O_2_S [M + H]^+^: 324.0550, found: 324.0551; IR ν_max_ (cm^−1^): 3110, 3067, 2937, 1543, 1519, 1488,1417, 1344, 1318, 1288, 1261, 1247, 1188, 1108, 1089, 865, 854, 838, 825, 757. Anal. calcd. for C_15_H_9_N_5_O_2_S: C, 55.72; H, 2.81; N, 21.66. Found: C, 55.70; H, 2.84; N, 21.63.

#### 3.1.2. General Procedure for the Preparation of Di(het)aryl-Substituted Azolo[1,5-*a*]pyrimidines **7a**,**b** (Figure 2)

The reduction of nitro derivatives to anilines was carried out according to the standard procedure. Compounds **4c**,**d** (0.2 g) were dissolved in a heated mixture of THF and ethanol (1:1); Raney nickel was added, followed by dropwise addition of hydrazine hydrate (1–2 mL). The reduction was carried out until the complete disappearance of the starting compound, testing by TLC (eluent: chloroform/ethanol, 50:1). The following products were crystallized from ethanol:

4,4′-(Imidazo[1,2-*a*]pyrimidine-5,7-diyl)dianiline **7a**: light yellow solid; mp = 169–171 °C. Yield: 145 mg (87%). ^1^H NMR (400 MHz, DMSO-*d*_6_) δ, 7.99 (m (AA’BB’), 2 H, phenyl), 7.79 (d, 1 H, *J* = 1.4 Hz, imidazopyrimidine), 7.60 (d, 1 H, *J* = 1.4 Hz, imidazopyrimidine), 7.57 (m (AA’BB’), 2 H, phenyl), 7.34 (s, 1 H, CH, imidazopyrimidine), 6.75 (m (AA’BB’), 2 H, phenyl), 6.66 (m (AA’BB’), 2 H, phenyl), 5.82 (s, 2 H, NH_2_) 5.68 (s, 2 H, NH_2_) (Appendix A). ^13^C NMR (101 MHz, DMSO-*d*_6_) δ 156.22, 151.36, 151.21, 149.79, 146.33, 134.06, 129.32, 128.44, 124.05, 118.50, 113.56, 113.50, 108.78, 102.65 (Appendix A). HRMS(ESI): calculated for C_18_H_16_N_5_ [M + H]^+^: 302.1400, found: 302.1401; IR ν_max_ (cm^−1^): 3444, 3334, 3204, 3029, 1594, 1530, 1498, 1471, 1385, 1293, 1273, 1241, 1178, 1139, 825, 708, 684. Anal. calcd. for C_18_H_15_N_5_: C, 71.74; H, 5.02; N, 23.24. Found: C, 71.71; H, 5.06; N, 23.20.

4-(7-(Thiophen-2-yl)-[1,2,4]triazolo[1,5-*a*]pyrimidin-5-yl)aniline **7b**: light yellow solid; mp = 211–214 °C. Yield: 150 mg (83%). ^1^H NMR (400 MHz, DMSO-*d*_6_) δ, 8.72 (dd, 1 H, *J* = 1.0, 4.0 Hz, thienyl), 8.65 (s, 1 H, CH, triazolopyrimidine), 8.27 (s, 1 H, CH, triazolopyrimidine), 8.16 (m (AA’BB’), 2 H, phenyl), 8.15 (dd, 1 H, *J* = 1.0, 4.9 Hz, thienyl), 8.43 (dd, 1 H, *J* = 4.0, 4.9 Hz, thienyl), 6.71 (m (AA’BB’), 2 H, phenyl), 5.92 (s, 2 H, NH_2_) (Appendix A). ^13^C NMR (101 MHz, DMSO-*d*_6_) δ, 160.51, 155.70, 155.25, 152.23, 140.36, 134.59, 132.61, 130.28, 129.33, 127.95, 122.82, 113.45, 101.46 (Appendix A). HRMS(ESI): calculated for C_15_H_12_N_5_S [M + H]^+^: 294.0808, found: 294.0806; IR ν_max_ (cm^−1^): 3559, 3421, 3326, 3215, 3099, 1545, 1504, 1383, 1317, 1302, 1245, 1197, 1178, 825, 773, 727. Anal. calcd. for C_15_H_11_N_5_S: C, 61.42; H, 3.78; N, 23.87. Found: C, 61.45; H, 3.76; N, 23.88.

#### 3.1.3. General Procedure for the Preparation of 7-Aryl-5-methyl-4,7-dihydro-[1,2,4]triazolo[1,5-*a*]pyrimidine-6-carboxylates **10a**,**b** (Figure 2)

A mixture of 20 mL of anhydrous CH_2_Cl_2_ and 0.6 mmol of Boc-glycine was heated to 45 °C; then, 10.1 mmol of TBTU and 0.9 mmol of DIPEA were added and stirred for 25–30 min. After that, 1 mmol of compound **8a** or **8b** was added, and the mixture was continuously stirred for 48 h at 45 °C. Further, the reaction mixture was kept without heating or stirring for another 48 h. The reaction mixture was filtered through thin layer of silicaand washed with CH_2_Cl_2_/ethyl acetate (1:1). To remove the protective group, intermediate compound **9a** or **9b** was dissolved in 1 mL of CF_3_COOH and left in air until the acid completely evaporated. The precipitate was washed with NaHCO_3_ solution and water, then dried in air. If necessary, product **10** was crystallized from THF.

Ethyl 7-(3-(2-aminoacetamido)phenyl)-5-methyl-4,7-dihydro-[1,2,4]triazolo[1,5-*a*]pyrimi-dine-6-carboxylate **10a**: yellow powder; mp = 270–272 °C. Yield: 320 mg (63%). ^1^H NMR (400 MHz, DMSO-*d*_6_) *δ*, 10.84 (br.s, 1H, NH), 10.46 (br.s, 1H, NH), 7.88 (br.s, 2H, NH_2_), 7.65 (s, 1H, CH), 7.50–7.48 (m, 2H, Ar), 7.28 (t, *J* = 7.8 Hz, 1H, Ar), 7.00 (d, *J* = 7.7 Hz, 1H, Ar), 6.24 (s, 1H, CH), 4.00–3.93 (m, 2H, CH_2_, OEt), 3.71 (s, 2H, CH_2_), 2.42 (s, 3H, Me), 1.06 (t, *J* = 7.1 Hz, 3H, CH_3_, OEt) (Appendix A). ^13^C NMR (101 MHz, DMSO-*d*_6_) *δ*, 165.4, 165.2, 157.7, 150.1, 146.4, 141.3, 140.1, 126.9, 122.4, 119.3, 116.5, 97.1, 59.5, 59.1, 41.2, 18.4, 14.0 (Appendix A). FT-IR (neat) ν_max_ (cm^−1^): 2947, 1644, 1596, 1509, 1289, 1100. Anal. calcd. for C_17_H_20_N_6_O_3_: C 57.29; H 5.66; N 23.58. Found: C 57.27; H 5.16; N 23.59.

Ethyl 7-(4-(2-aminoacetamido)phenyl)-5-methyl-4,7-dihydro-[1,2,4]triazolo[1,5-*a*]pyrimi-dine-6-carboxy-late **10b**: yellow powder; mp = 270–272 °C. Yield: 320 mg (63%). ^1^H NMR (400 MHz, DMSO-d_6_) *δ*, 10.80 (br.s, 1H, NH), 10.46 (br.s, 1H, NH), 7.94 (br.s, 2H, NH_2_), 7.65 (s, 1H, CH), 7.51 (d (AA’BB’), *J* = 8.4 Hz, 2H, Ar), 7.21 (d, (AA’BB’), *J* = 8.4 Hz, 2H, Ar), 6.23 (s, 1H, CH), 4.00–3.92 (m, 2H, CH_2_, OEt), 3.73 (s, 2H, CH_2_, 2.41 (s, 3H, Me), 1.04 (t, *J* = 7.1 Hz, 3H, CH_3_, OEt) (Appendix A). ^13^C NMR (101 MHz, DMSO-*d*_6_) *δ*, 165.2, 165.1, 157.8, 150.1, 146.7, 137.8, 137.6, 127.7, 119.1, 97.1, 41.2, 59.3, 59.0, 18.4, 13.9 (Appendix A). FT-IR (neat) ν_max_ (cm^−1^): 2947, 1644, 1596, 1509, 1289, 1100. Anal. calcd. for C_17_H_20_N_6_O_3_: C 57.29; H 5.66; N 23.58. Found: C 57.12; H 5.69; N 23.60.

#### 3.1.4. The General Synthesis of 6-Nitro-7-methoxy-4,7-dihydro-azolyl-4 guanidates **12a**,**b**, **14** (Figure 3)

A 1 M solution of free guanidine in methanol (1 mL) was added to a solution of 1 mmol of the starting 6-nitroazolopyrimidine in 5 mL of methanol. The solution was stirred at 60 °C for 10 min; the precipitated product was separated by filtration, washed with cold methanol, and dried in a vacuum desiccator.

2-Trifluoromethyl-6-nitro-7-methoxy-4,7-dihydro[1,5-*a*]pyrimidyl-4 guanidate **12a**: yellow powder; mp = 167–168 °C. Yield: 240 mg (74%). ^1^H NMR (400 MHz, DMSO-*d*_6_) δ, 8.51 (s, 1H, H^5^), 6.97 (br.s, 6H), 6.70 (s, 1H, H^7^), 3.26 s, 3H, CH_3_) (Appendix A). ^13^C NMR (126 MHz, DMSO-*d*_6_) δ, 157.95, 157.15, 151.57, 151.25 (q), 119.72 (q), 116.89, 83.24, 55.31 (Appendix A). IR ν_max_ (cm^−1^): 1000 (C-F); 1670, 1686 (C=N), 3141, 3359 (NH). Anal. calcd. for C_8_H_11_N_8_O_3_F_3_C: C 29.63; H 3.40; N 34.22. Found: 29.50; H 3.46; N 34.44.

2-Cyclopentylthio-6-nitro-7-methoxy-4,7-dihydro[1,5-*a*]pyrimidyl-4 guanidate **12b**: yellow powder; mp = 164–166 °C. Yield: 275 mg (77%). ^1^H NMR (500 MHz, DMSO-*d*_6_) δ, 8.43 (s, 1H, H^5^), 6.59 (br.s, 6H), 6.51 (s, 1H, H^7^), 3.83–3.78 (m, 1H), 3.21 (c, 3H, CH_3_), 2.13–2.06 (m, 2H), 1,70–1.55 (2m, 6H) (Appendix A). ^13^C NMR (126 MHz, DMSO-*d*_6_) δ, 158.66, 157.86, 156.54, 150.95, 116.36, 82.52, 54.90, 43.83, 33.51, 33.23, 24.31, 24.28 (Appendix A). IR ν_max_ (cm^−1^): 1682 (C=N), 3107, 3325 (NH). Anal. calcd. for C_12_H_20_N_8_O_3_S: C 40.45; H 5.62; N 31.46. Found: C 40.59; H 5.57; N 31.30.

6-Nitro-7-hydroxy-4,7-dihydro-1,2,4-triazolo[5,1-*c*]triazinyl-4 guanidate **14**: burgundy powder; mp > 300 °C. Yield: 165 mg (68%). ^1^H NMR (500 MHz, DMSO-*d*_6_) δ, 7.80 (s, 1H, H^2^), 7.07 (br.s, 6H), 6.91 (s, 1H, H^7^) (Appendix A). ^13^C NMR (126 MHz, DMSO-*d*_6_), δ, 157.91, 154.47, 150.99, 140.66, 72.06 (Appendix A). IR ν_max_ (cm^−1^): 1680 (C=N), 3107, 3318 (NH). Anal. calcd. for C_5_H_9_N_9_O_3_: C 24.50; H 3.79; N 51.71. Found: C 24.69; H 3.70; N 51.85.

#### 3.1.5. The General Synthetic Method of 6-Guanidino-3-alkylthio-[1,2,4]triazolo[4,3-*b*][1,2,4,5]tetrazines **16a**–**c** (Figure 5)

1.0 mL (C = 1.0 mmol·mL^−1^) of freshly prepared solution of free guanidine in methanol was added dropwise to a mixture of 1 mmol of the corresponding [1,2,4]triazolo[1,2,4,5]tetrazine in 10 mL of acetonitrile while stirring on a magnetic stirrer. The precipitate of 6-guanidino-3-alkylthio-[1,2,4]triazolo[4,3-*b*][1,2,4,5]tetrazine that formed after 15 min was filtered, washed on the filter with acetonitrile, and crystallized from water or acetonitrile.

6-Guanidino-3-octylthio-[1,2,4]triazolo[4,3-*b*][1,2,4,5]tetrazine **16a**: yellow powder; mp > 260 °C. Yield: 266 mg (82%). ^1^H NMR (400 MHz, DMSO-*d*_6_) *δ*, 0.83 (t, 3H, CH_3_, J = 7.0 Hz); 1.23–1.26 (m, 8 H, 4 CH_2_); 1.36–1.39 (m, 2 H, CH_2_); 1.63–1.69 (m, 2 H, CH_2_); 3.22–3.25 (t, 2 H, SCH_2_, *J* = 7.0 Hz); 7.20 (br.s, 4H guanidino) (Appendix A). ^13^C NMR (101 MHz, DMSO-*d*_6_) *δ*, 13.9, 22.0, 27.8, 28.3, 28.5, 29.4, 31.1, 31.4, 142.3, 149.9, 158.4, 159.8 (Appendix A). IR ν_max_ (cm^−1^): 1639 (C=NH, guanidino), 3414,3447 (N-H, guanidino). Anal. calcd. for C_12_H_21_N_9_S: C, 44.56; H, 6.54; N, 38.98. Found: C, 44.26; H, 6.70; N, 39.23.

6-Guanidino-3-cyclopentylthio-[1,2,4]triazolo[4,3-*b*][1,2,4,5]tetrazine **16b**: orange powder; mp = 252–253 °C. Yield: 142 mg (51%). ^1^H NMR (400 MHz, DMSO-*d*_6_) *δ*, 1.58–1.67, 1.70–1.77, 2.05–2.12 (3 m, 8 H, 4 CH_2_); 3.96–4.01 (m, 1 H, CH, cyclopentyl); 7.24 (br.s, 4H guanidino) (Appendix A). ^13^C NMR (101 MHz, DMSO-*d*_6_) *δ*, 24.1, 33.4, 45.4, 142.1, 149.8, 158.5, 159.8. IR ν_max_ (cm^−1^): 1644 (C=NH, guanidino), 3381,3416 (N-H, guanidino) (Appendix A). Anal. calcd. for C_9_H_13_N_9_S: C, 38.70; H, 4.69; N, 45.13. Found: C, 38.31; H, 4.70; N, 45.23.

6-Guanidino-3-(3-fluorophenylmethylthio[1,2,4]triazolo[4,3-*b*][1,2,4,5]tetrazine **16c**: yellow powder; mp = 235–237 °C. Yield: 280 mg (87%). ^1^H NMR (400 MHz, DMSO-*d*_6_) *δ*, 4.49 (c, 2 H, SCH_2_); 7.05–7.33 (m, 8H, guanidino, Ph) (Appendix A). ^13^C NMR (101 MHz, DMSO-*d*_6_) *δ*, 34.7, 114.3 (d, J =20.9), 125.6 (d, *J* = 21.8 Hz), 124.9 (d, *J* =2.6 Hz), 130.30 (d, *J*= 8.4 Hz), 139.9 (d, *J* = 7.7 Hz), 141.3, 149.9, 158.4, 159.7, 161.8 (d, *J* = 243.8 Hz) (Appendix A). IR-spectra, IR ν_max_ (cm^−1^): 1645 (C=NH, guanidino), 3382, 3424 (N-H, guanidino). Anal. calcd. for C_11_H_10_FN_9_S: C, 41.37; H, 3.16; N, 39.48. Found: C, 41.62; H, 3.30; N, 39.38.

### 3.2. Biology

#### 3.2.1. Animals

All animal procedures were carried out under the generally accepted ethical standards for animal testing adopted by the European Convention for the Protection of Vertebrate Animals used for Experimental and Other Scientific Purposes (1986) and taking into account the International Recommendations of the European Convention for the Protection of Vertebrate Animals Used for Experimental Research (1997). This study was approved by the Local Ethics Committee of Volgograd State Medical University (registration no. IRB 00005839 IORG 0004900, OHRP), certificate no. 2021/056, 15 June 2021. All sections of this study adhered to the ARRIVE Guidelines for reporting animal research [33]. The experiments were carried out on 10 male Chinchilla rabbits weighing 3.0–3.5 kg and 60 outbred albino male rats weighing 250–270 g. Animals were kept under standard vivarium conditions (22–24 °C, 40–50% humidity, ambient light) during the study. The control and experimental groups included 10 samples each.

#### 3.2.2. In Vitro Anticoagulant Assay

The study was performed on platelet-poor plasma (PPP) stabilized with a 3.8% sodium citrate solution in a ratio of 9:1 according to the method described in [34]. Dabigatran etexilate (Boehringer Ingelheim Pharma GmbH and Co., Ingelheim am Rhein, Germany) was used as the reference drug. Test compounds and the reference drug were evaluated at a concentration of 100 μM. Effect on rabbit blood coagulograms in vitro was determined chronometrically with a SOLAR hemocoagulometer (Minsk, Belarus) using commercial kits (Technology-Standard, Barnaul, Russia) as per the manufacturer’s instructions. The following parameters were determined: activated partial thromboplastin time, thrombin time, and prothrombin time.

Hypercytokinemia conditions were modeled by incubation of whole blood with S. typhimurium LPS (Sigma Aldrich, St. Louis, MO, USA) at a final concentration of 20 ng/mL and subsequent preparation of PPP. Compounds that showed high dose-dependent prolongation of thrombin time with LPS treatment and without were assessed for IC_50_ values using the regression analysis method in the Microsoft Excel 2007 program (Microsoft Corporation, Albuquerque, NM, USA).

#### 3.2.3. Ecarin Clotting Time

The ecarin clotting time was designed for the determination of antithrombin anticoagulant activity in platelet-poor plasma [35]. Venous blood was collected into a 15 mL Falcon type plastic tube with 3.8% sodium citrate solution (0.5 mL of sodium citrate per 5 mL of blood), centrifuged on a CM-6M centrifuge (Elmi, Riga, Latvia) at room temperature from 18 to 25 °C for 15 min at 3000 rpm (1200× *g*). The obtained plasma was withdrawn using an automatic pipette and transferred to a separate clean, dry plastic tube. After that, the obtained plasma was recentrifuged under the same conditions and the upper third of the plasma was taken for analysis.

The ecarin stock solution (STA-ECT, Stago, Düsseldorf, Germany) was prepared according to the manufacturer’s instructions and diluted to 4 units/mL with HEPES-buffered saline (0.2 M) containing 0.025 M calcium chloride.

The most active compound was selected as the study sample and the anticoagulant drug dabigatran etexilate was chosen as the comparison drug. The tested samples were investigated in the concentration range from 100 to 1 μM for IC_50_ determination.

Prepared reagents and platelet-poor plasma were heated to 37 °C. The assay was performed on a Solar CGL2120 coagulometer (Belarus), adding 50 µL of the reagent ecarin to 100 µL of PPP and recording the clotting time. As for substances exhibiting antithrombin action, the test was performed by adding 10 µL of the compound to 100 µL of PPP followed by an incubation for 5 min, after which 50 µL of the reagent ecarin was added and the clotting time was recorded.

#### 3.2.4. Anticoagulant Assay in Animals

The most active compound was studied on male rats in vivo, in a single intragastric administration in a volume of no more than 2 mL. Distilled water was used as a solvent. In all experiments, control animals were injected with an equivalent volume of a vehicle. The reference drug dabigatran etexilate was administered to rats 2 h before the study at a 12 mg/kg dose (equivalent to the human dose, taking into account the interspecies conversion factor). Compound **5d** was administered to rats 2 h before the study in a 5.5 mg/kg dose, equimolar to the dose of dabigatran etexilate.

Blood was taken from the inferior vena cava of rats anesthetized with 50 mg/kg pentobarbital intraperitoneally. To stabilize the blood, a 3.8% aqueous solution of sodium citrate (pH 6.0) was used in a ratio of 9:1. Coagulogram parameters of platelet-poor plasma were measured with a SOLAR coagulometer according to the methods described above.

Hypercytokinemia conditions were modeled by an intravenous injection of S. typhimurium LPS (Sigma Aldrich, St. Louis, MO, USA) at a final dose of 2 mg/kg [36] into the tail vein of the rats 2 h before the administration of the substances. The evaluation of anticoagulant activity was conducted according to the methods described above.

#### 3.2.5. Statistical Analysis

Biological data were analyzed with one-way ANOVA using Bonferroni’s multiple comparison correction in the Microsoft Excel 2007 spreadsheet editor, STATISTICA 5.0 (StatSoft, Inc., Tulsa, OK, USA) for Windows, and Prism 5.0 (GraphPad Inc., San Diego, CA, USA). Data were presented as M + m, where m is a SEM. Changes were statistically significant if *p* < 0.05.

The calculation of EC_50_ (effective concentration that prolongs the time of clot formation by 50%) was performed using linear regression analysis.

## 4. Conclusions

Venous thromboembolism (VTE) is a serious clinical problem, associated with significant morbidity and mortality, including in viral and bacterial infections [37]. The process of thrombosis prevention is a delicate balance between the inhibition of hypercoagulability, stasis, and damage of the vascular wall. The basis for protecting the body from the effects of external and internal environments is inflammation and blood clotting, known as immunocoagulation, which is a part of innate immunity and can serve as the first line of defense against infection [38]. However, uncontrolled inflammation is associated with a pronounced reaction of the immune system, which leads to a hypercoagulable state, which is manifested by the development of macro- and microvascular thrombosis of the venous and arterial beds [39]. It is known that coagulation can be activated by extrinsic and intrinsic pathways, leading to the formation of fibrin. Preclinical and clinical studies have confirmed the pathological role of tissue factor, the initiator of the extrinsic pathway, in the development of endotoxemia [40]. It has been experimentally shown that exogenous lipopolysaccharide (LPS) can induce the expression and release of tissue factor on the cell surface and lead to septic death in mice [41].

In addition, hypercytokinemia, observed in sepsis, not only causes the activation of coagulation factors, but also suppresses anticoagulant pathways, such as the antithrombin system, activated protein C, and tissue factor inhibitor, thus leading to disseminated intravascular coagulation and fibrin deposition in the blood vessels and tissues. So, the interaction between increasing immune dysfunction and abnormal blood clotting is a major event favoring the complications of sepsis and multiple organ failure in humans and should be strategically targeted for therapeutic purposes [42]. Therefore, the use of anticoagulants for viral and bacterial infections helps prevent blood clots.

Direct thrombin inhibitors are important anticoagulant drugs, acting through the inhibition of the common pathway via factor IIa. These drugs can selectively bind to the active site of thrombin, inhibit thrombin activity, and exhibit a strong action and high specificity, and are important for the clinical treatment of thrombus diseases. Direct thrombin inhibitors are more efficient than other anticoagulants (such as heparin and warfarin), due to their higher capacity for inhibiting both free and bound thrombin, having a relatively safe pharmacological profile, and lacking the need for cofactors [43]. Therefore, the search for new direct thrombin inhibitors is currently in a great demand. A series of 27 novel azoloazine derivatives were evaluated in vitro for their anticoagulant properties by estimating the coagulogram parameters of rabbit blood. As a result, 26 compounds proved to prolong the thrombin time relative to the control, but to a lesser extent than the reference drug dabigatran etexilate. None of the studied compounds affected the prothrombin time.

It was established that the most active compound, **5d**, had the highest ability to prolong the thrombin time, exceeding the reference drug dabigatran etexilate by 13.0 times, but it did not affect the prothrombin time, thus indicating its ability to inhibit thrombin.

At present, the gold standard for the study of direct oral anticoagulants is the ecarin clotting time, wherein ecarin maximizes thrombin activity and clotting time is evaluated to assess direct thrombin-inhibiting anticoagulation capability. Ecarin is a metalloprotease enzyme derived from the venom of the Indian saw-scaled viper. Ecarin cleaves prothrombin’s Arg320 peptide bond at the alanine–arginine–aspartic acid peptide motif, yielding the thrombin intermediate meizothrombin. Meizothrombin has up to 97% thrombin activity (compared with 10% without ecarin), and it converts fibrinogen to fibrin much more efficiently. Because direct thrombin inhibitors can inhibit not only thrombin, but also meizothrombin, ecarin-based assays are quite appropriate to estimate anticoagulation capability under optimal conditions, compared with conventional coagulation analyses [44]. Therefore, as the next step of our study, the effect of compound **5d** on ecarin time was estimated. The results of the ecarin time study show that the direct antithrombin effect of the compound under investigation is comparable with the reference drug dabigatran etexilate in terms of EC_50_. This study demonstrates that the novel compound **5d** (dimethyl-4,5-dihydro-[1,2,4]triazolo[1,5-*a*]pyrimidine) exhibits direct antithrombin anticoagulant activity. Thus, among the class of compounds studied, substituted triazolopyrimidines and their hydrogenated analogues should be recognized as the most promising for further research on anticoagulants.

The antithrombin activity of compound **5d** was confirmed using LPS-treated rabbit blood to mimic the conditions of cytokine release syndrome. Under conditions of hypercytokinemia, both compound **5d** and dabigatran etexilate showed higher activity than those obtained for the blood of intact animals, and the EC_50_ values of their antithrombin activity proved to be comparable.

Compound **5d** was also evaluated in animals via a single intragastric administration of **5d** to rats in doses equimolar to those of dabigatran etexilate with and without hypercytokinemia conditions. Compound **5d** and dabigatran etexilate proved to increase the thrombin time for normal animals by 5.6 and 10.5 times, respectively. In animals under conditions of a systemic inflammation reaction, the antithrombin activity of dabigatran etexilate did not change. The antithrombin effect of compound **5d** was twice as pronounced than in the absence of a systemic inflammation reaction.

Thus, the enhancement of the anticoagulant effect of compound **5d** under conditions of a systemic inflammatory response indicates the presence of anti-inflammatory activity. Consequently, this compound can make a significant contribution by influencing the pathogenetic links of immunocoagulation and thereby reducing the risk of thrombosis, including in conditions of viral and bacterial infections. It is difficult to predict the structural modifications that could lead to improved activity of compound **5d**, but there should probably be a substituent on the thiophene rings.

## Data Availability

Data are contained within the article.

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
