# Peer review of "Novel Substituted Azoloazines with Anticoagulant Activity"

_ijms, 2023, doi:10.3390/ijms242115581_

Round 1
Reviewer 1 Report
The manuscript on "Novel Substituted Azoloazines with Anticoagulant Activity" by A. A. Spasov et al. is an experimental investigation of several new compounds comparable in their anticoagulant properties to the well-established medication, dabigatran etexilate (Pradaxa). The study is well designed, there are control experiments present, including Pradaxa as the reference. Description of the procedures is adequate, the supporting information contains spectral evidence for the synthesis. I have only two small comments to the overall well-written manuscript.
1) Please mention (close to line 62) that dabigatran etexilate is also an azoloazine; it might be beneficial for the reader to add its structure in the Figure 1.
2) Figure 2: the dashed lines are not explained. Are they just linear regression results? Seemingly from three points only, therefore their quality might be questioned. Maybe they can be safely removed.
End of the reviewer remarks.
Author Response
We are grateful to the respected reviewers for their constructive comments. Please, find below the detailed point-by-point response to your concerns. We believe that manuscript's quality is improved thanks to your suggestions.
|
Comments by Reviewer 1: |
Response |
|
The manuscript on "Novel Substituted Azoloazines with Anticoagulant Activity" by A. A. Spasov et al. is an experimental investigation of several new compounds comparable in their anticoagulant properties to the well-established medication, dabigatran etexilate (Pradaxa). The study is well designed, there are control experiments present, including Pradaxa as the reference. Description of the procedures is adequate, the supporting information contains spectral evidence for the synthesis. I have only two small comments to the overall well-written manuscript. 1) Please mention (close to line 62) that dabigatran etexilate is also an azoloazine; it might be beneficial for the reader to add its structure in the Figure 1. 2) Figure 2: the dashed lines are not explained. Are they just linear regression results? Seemingly from three points only, therefore their quality might be questioned. Maybe they can be safely removed. |
Thank you for the suggestion and remarks. Corrections were made in the text.
1) In the line 62 we mentioned that dabigatran etexilate is also an azoloazine and added its structure in the Figure 1. 2)At the Figure 2 the dashed lines were removed. |
Reviewer 2 Report
Alexander A. Spasov et al reported the synthesis of novel substituted azoloazines and evaluation of in vitro and in vivo anticoagulant activity. My comments are as follows.
1. The authors are suggested to describe the numbers of animal used or in vitro studies been conducted (duplicate or triplicate) in the figure/table legend.
2. In Table 3, the authors are suggested to discuss the reason that on dose-dependent effect for both control compound and 5d.
3. The authors are suggested to discuss SAR and future direction of structure modification to improve activities.
4. The authors are suggested to discuss the possible mechanisms that under conditions of a systemic inflammatory reaction, compound 5d had shown a more pronounced antithrombin effect.
Proof read by native English speaker is recommended.
Author Response
We are grateful to the respected reviewers for their constructive comments. Please, find below the detailed point-by-point response to your concerns. We believe that manuscript's quality is improved thanks to your suggestions.
|
Comments by Reviewer 2: |
Response |
|
Alexander A. Spasov et al reported the synthesis of novel substituted azoloazines and evaluation of in vitro and in vivo anticoagulant activity. My comments are as follows. 1. 1) The authors are suggested to describe the numbers of animal used or in vitro studies been conducted (duplicate or triplicate) in the figure/table legend. 2. 2) In Table 3, the authors are suggested to discuss the reason that on dose-dependent effect for both control compound and 5d. 3. 3) The authors are suggested to discuss SAR and future direction of structure modification to improve activities. 4. 4) The authors are suggested to discuss the possible mechanisms that under conditions of a systemic inflammatory reaction, compound 5d had shown a more pronounced antithrombin effect. |
Thank you for the suggestion and remarks Corrections were made in the text. 1)We mentioned the numbers of animal used in studies in the name to the table and in its notes.
2) By mistake, the data in the table was added incorrectly, that’s why presentation of data was wrong, everything was corrected. 3) Information added to Conclusions chapter 4) Information added to Conclusions chapter.
|